# Clusters of Hydroxyl-Functionalized Cations Stabilized by Cooperative Hydrogen Bonds: The Role of Polarizability and Alkyl Chain Length

**DOI:** 10.3390/molecules25214972

**Published:** 2020-10-27

**Authors:** Jule K. Philipp, Ralf Ludwig

**Affiliations:** 1Institut für Chemie, Abteilung Physikalische und Theoretische Chemie, Universität Rostock, 18059 Rostock, Germany; jule.philipp@uni-rostock.de; 2Department Life, Light & Matter, Universität Rostock, Albert-Einstein-Str. 25, 18059 Rostock, Germany; 3Leibniz-Institut für Katalyse an der Universität Rostock e.V., Albert-Einstein-Str. 29a, 18059 Rostock, Germany

**Keywords:** like charge attraction, ionic liquids, hydrogen bonding, spectroscopic descriptors, DFT calculations

## Abstract

We explore quantum chemical calculations for studying clusters of hydroxyl-functionalized cations kinetically stabilized by hydrogen bonding despite strongly repulsive electrostatic forces. In a comprehensive study, we calculate clusters of ammonium, piperidinium, pyrrolidinium, imidazolium, pyridinium, and imidazolium cations, which are prominent constituents of ionic liquids. All cations are decorated with hydroxy-alkyl chains allowing H-bond formation between ions of like charge. The cluster topologies comprise linear and cyclic clusters up to the size of hexamers. The ring structures exhibit cooperative hydrogen bonds opposing the repulsive Coulomb forces and leading to kinetic stability of the clusters. We discuss the importance of hydrogen bonding and dispersion forces for the stability of the differently sized clusters. We find the largest clusters when hydrogen bonding is maximized in cyclic topologies and dispersion interaction is properly taken into account. The kinetic stability of the clusters with short-chained cations is studied for the different types of cations ranging from hard to polarizable or exhibiting additional functional groups such as the acidic C(2)-H position in the imidazolium-based cation. Increasing the alkyl chain length, the cation effect diminishes and the kinetic stability is exclusively governed by the alkyl chain tether increasing the distance between the positively charged rings of the cations. With adding the counterion tetrafluoroborate (BF_4_^−^) to the cationic clusters, the binding energies immediately switch from strongly positive to strongly negative. In the neutral clusters, the OH functional groups of the cations can interact either with other cations or with the anions. The hexamer cluster with the cyclic H-bond motive and “released” anions is almost as stable as the hexamer built by H-bonded ion pairs exclusively, which is in accord with recent IR spectra of similar ionic liquids detecting both types of hydrogen bonding. For the cationic and neutral clusters, we discuss geometric and spectroscopic properties as sensitive probes of opposite- and like-charge interaction. Finally, we show that NMR proton chemical shifts and deuteron quadrupole coupling constants can be related to each other, allowing to predict properties which are not easily accessible by experiment.

## 1. Introduction

Opposite charges attract and like charges repel each other. The common wisdom is known from Coulomb’s law for the electrostatic interaction between charged particles. The pairing between opposite-charged ions is also a well-established concept in chemistry and plays an important role for reactions in solution, macromolecular catalysis, biochemical hydrolysis, and protein stability [1,2,3]. In contrast, attractive interaction between ions of like charge seems to be a counterintuitive phenomenon. However, like-charge attraction was reported for aqueous salt solutions [4], guanidinium ions in water [5], micellization of tetraalkylammonium surfactants [6], and biomolecules like oligopeptides and DNA [7,8]. Usually, large-scale structures or assemblies are needed for stabilizing the interaction between like-charged particles. In solution, mediating solvent molecules such as water are required. For ionic liquids (ILs), which consist of ions solely, attractive interaction was dominantly reported between cation and anion due to strong Coulomb forces [9,10,11,12,13,14,15,16,17]. Meanwhile, even for ionic liquids, increasing evidence for attractive interactions between ions of like charge is reported. In hydroxyl-functionalized ILs, we recently observed two types of hydrogen bonds (H-bonds): normal H-bonds between cation and anion, further enhanced by attractive Coulomb forces, and elusive H-bonds between two or more cations leading to cluster formation of like-charged ions that are supposed to be much weaker due to the repulsive Coulomb force [18,19,20,21,22]. Despite this expectation, the hydrogen bonds in cationic clusters are evidently stronger than the ones in ion pairs as shown by stronger redshifted OH vibrational bands in IR spectra [23,24]. Consequently, we detected cationic clusters in the bulk liquid as well as in the gas phase by spectroscopic methods [25,26,27,28,29,30,31]. We observed significantly enhanced cationic cluster formation for hydroxyl-functionalized ILs (*n*-hydroxyalkyl)-pyridinium bis(trifluoromethanesulfonyl)imide [HO-(CH_2_)*_x_*-Py][NTf_2_]. These ILs comprise polarizable pyridinium cations, weakly interacting anions, and sufficiently long hydroxyl alkyl chains (*n* = 4) at the cation for tethering the positively charged pyridinium ring away from the hydroxy group, allowing hydrogen bonding between the OH groups of the like-charged ions. The question arises whether pure cationic clusters [HO-(CH_2_)*_x_*-Py^+^]*_n_* can exist without the mediating “solvent effects” by polarizable cations or weakly interacting counterions. Recently, we explored quantum chemical calculations for studying the stability of cationic dimers [HO-(CH_2_)*_x_*-Py^+^]_2_ [32]. Although we could significantly decrease the Coulomb repulsion, very long hydroxy-alkyl chains with *x* = 15 were needed before meta-stable dimers changed into stable dimers, wherein the Coulomb repulsion is fully compensated by the attractive hydrogen bond. Depending on the alkyl chain length, we could also calculate larger cationic clusters up to cyclic pentamers and hexamers [33]. Taking dispersion effects into account, we found minimum structures for the cyclic pentamer [HO-(CH_2_)_2_-Py^+^]_5_ and even for the cyclic hexamer [HO-(CH_2_)_3_-Py^+^]_6_ due to slightly longer hydroxyalkyl chains in the latter (see Figure 1a,b). The structural motifs of the calculated cationic clusters very much resemble the cyclic pentamers and hexamers as observed experimentally for water and alcohols [34,35,36,37,38].

All these findings support the concept of so-called “anti-electrostatic hydrogen bonds” (AEHBs) recently reported by Weinhold and Klein [39]. Unusual kinetic stability for cation–cation and anion–anion complexes despite strongly repulsive electrostatic forces between the like-charged ions challenges the earlier used electrostatic model for hydrogen bonds (HBs) and supports the current concept that considers the importance of covalency in hydrogen bonding [40,41,42,43,44].

In this comprehensive study, we calculate clusters of hydroxyl-functionalized cations starting from linear dimers up to the size of cyclic hexamers. We focus on the delicate balance of Coulomb interaction, hydrogen bonding, and dispersion forces for the kinetic stability of these clusters of like-charged ions. We show that cooperative hydrogen bonding and dispersion interaction prevent the clusters from “Coulomb explosion” and even allow the formation of kinetically stable cyclic hexamers with net charge *Q* = +6*e*. For the hydroxyethyl-chained cations, we isolate the role of polarizability for the cluster stability. Then, we investigate the influence of increasing hydroxy-alkyl chain lengths in [HO-(CH_2_)*_x_*-Py^+^] cations with *x* = 2–6 for preferential cationic cluster formation. Neutralizing the cationic clusters by adding the same number of counterions allows studying the competition between complexes including the cationic HB-motifs and those solely formed by ion pairs. Moreover, we calculated spectroscopic descriptors such as NMR proton chemical shifts *δ*^1^H and deuteron quadrupole coupling constants (DQCCs) *χ_D_* for characterizing hydrogen bonding in the cationic and neutral complexes. Here, we show that the calculated geometries and spectroscopic properties of the ionic or neutral, gas-phase-like clusters are similar to those observed in the liquid phase, suggesting that these structural motifs are really present in ionic liquids.

## 2. Density Functional Theory (DFT) Calculations of Cationic and Neutral Clusters

We calculated clusters of the hydroxyl-functionalized cations 1-(*x*-hydroxyalkyl)pyridinium [HO-(CH_2_)*_x_*-Py^+^], 1-(*x*-hydroxyalkyl)methylpiperidinium [HO-(CH_2_)*_x_*-Pip^+^], 1-(*x*-hydroxyalkyl)-1,1,1-trimethylammonium [HO-(CH_2_)*_x_*-TMA^+^], 1-(*x*-hydroxyalkyl)methylpyrrolidinium [HO-(CH_2_)*_x_*-Pyrro^+^], and 1-(*x*-hydroxyalkyl)-3-methylimidazolium [HO-(CH_2_)*_x_*-MIm^+^] with varying chain length between *x* = 2–6 (see Figure 1). The topologies comprise cationic monomers (*n* = 1), linear dimers (*n* = 2), and cyclic clusters from trimers (*n* = 3) up to hexamers (*n* = 6) as exemplarily shown for cationic clusters [HO-(CH_2_)_4_-Py^+^]*_n_* with *n* = 1–6 in Figure 2. The hydroxy-alkyl groups of the cations form (c–c) hydrogen bonds among each other and promote the aggregation into highly charged cationic clusters exhibiting strong cooperativity. Finally, we calculated neutral complexes ([HO-(CH_2_)_2_-Py][BF_4_])*_n_* with *n* = 1–6 by adding tetrafluoroborate [BF_4_^−^] counter-ions to the cationic clusters. In principle, two topologies can be formed; they are shown in Figure 3 for hexameric clusters ([HO-(CH_2_)_2_-Py][BF_4_])_6_. In Figure 3a, we show the simple ion pair characterized by attractive Coulomb interaction and additional (c–a) hydrogen bonding between cation and anion. One type of hexamer is built upon these ion pairs solely (see Figure 3b), whereas the other includes only (c–c) bound cationic clusters with additional anions interacting with the positively charged rings of the cations. We observed both types of hydrogen bonds in infrared spectra, whereby (c–c) H-bonds are stronger than (c–a) H-bonds and are strongly favored with decreasing temperature [20,21,22,23].

For cationic and neutral clusters, we employed B3LYP/6–31+G* and B3LYP-D3/6–31+G* calculations performed with the Gaussian 09 program [45]. For calculating the clusters at the same level of theory, we used the well-balanced 6–31+G* Pople basis set. Including polarization as well as diffuse functions, this basis set is suitable for reasonably calculating hydrogen-bonded clusters of like-charged ions [20,21,22,23]. We used the relatively small 6–31+G* basis set for calculating all clusters at the same level of theory as well as for better comparison with earlier studies of molecular and ionic clusters [46,47,48]. We demonstrated that the salient properties of these clusters can be robustly calculated with both smaller and larger basis sets as long as Grimme’s D3 dispersion correction is considered [49,50,51]. This is demonstrated for the features of the largest cationic and neutral clusters found here, namely cyclic hexamers. All clusters were fully optimized followed by frequency calculations. The obtained vibrational frequencies were all positive, showing that we calculated at least local minimum structures.

## 3. NMR Proton Chemical Shifts and Deuteron Quadrupole Coupling Constants as H-Bond Sensors

We also calculated two spectroscopic descriptors: the hydroxyl proton NMR chemical shifts, *δ*^1^H, and the deuteron quadrupole coupling constants, *χ*_D_, which are both sensitive probes for hydrogen bonding [52,53,54]. Firstly, we calculated the NMR chemical shielding values for all hydroxyl protons in the geometry-optimized cationic and neutral clusters. We then subtracted the average chemical shielding value of a cluster from the corresponding monomer value, resulting in the hydroxyl proton chemical shifts, *δ*^1^H, for all cluster species.

Additionally, we calculated the deuteron quadrupole coupling constant (DQCC), *χ*_D_, for each deuteron present in cluster configurations. The DQCC describes the coupling between the nuclear quadrupole moment, *eQ*, and the principle component of the electric field gradient tensor, *eq*_zz_, at the deuteron nucleus. It could be shown that the relation between *χ*_D_ and *eq*_zz_ is given by the equation
(1)$$\chi_{D}\;=\;\left(\frac{eQeq_{zz}}{h}\right)\left(\mathrm{kHz}\right)\;=\;2.3496\;eQ\;\left(\frac{{\mathrm{fm}}^{2}}{e}\right)eq_{zz}\;\left(a.u.\right)$$
where the factor 2.3496 takes care of the units. In principle, the DQCC can be obtained by multiplying the calculated principle component of electric field gradient tensor, *eq*_zz_, of the OD hydroxyl groups in the cationic clusters with a calibrated nuclear quadrupole moment, *eQ*. The calibrated *eQ* is obtained by plotting the measured gas phase quadrupole coupling constants from microwave spectroscopy versus the calculated electric field gradients for small molecules, such as H_2_O, CH_3_OH, H_2_CO_2_, etc., as described by Huber et al. [55,56,57]. The slope gives a reasonable value of *eQ* = 295.5 fm^2^, which should be used for calculating DQCCs at the B3LYP-D3/6–31+G* level of theory. It could also be shown for this set of molecules that the principal axis of the deuteron electric field gradient is nearly axially symmetric and lies along the direction of the O-D bonds [58]. We show the differences of the calculated average *χ*_D_ values in the clusters *n* and the corresponding monomer (*n* = 1) values, finally resulting in negative *χ*_D_ values caused by hydrogen bonding.

## 4. Influence of Cation Polarizability on Energies, Geometries, and Spectroscopic Properties

We calculated clusters of the hydroxyethyl-functionalized cations [HO-(CH_2_)_2_-Py^+^], [HO-(CH_2_)_2_-Pip^+^], [HO-(CH_2_)_2_-TMA^+^], [HO-(CH_2_)_2_-Pyrro^+^], and [HO-(CH_2_)_2_-MIm^+^] with varying cluster size from two to six. In Figure 4, we exemplarily show the intermolecular energies per species in the cationic clusters for [HO-(CH_2_)_2_-TMA^+^]*_n_*, [HO-(CH_2_)_2_-Py^+^]*_n_*, and [HO-(CH_2_)_2_-MIm^+^]*_n_* calculated with and without explicitly considering dispersion interaction using Grimme’s D3 correction [49,50,51]. We observe that for the short-chained hydroxyethyl cations [HO-(CH_2_)_2_Cat^+^]*_n_*, the polarizability is relevant for the formation of larger sized meta-stable clusters. Non-polarizable cations such as hydroxyethyl ammonium only allow the formation of linear dimers, whereas polarizable cations such as hydroxyethyl pyridinium and hydroxyethyl methyl-imidazolium show meta-stable cyclic tetramers. If we take dispersion forces explicitly into account, the cluster sizes increase to cyclic tetramers in the first and to cyclic pentamers in the latter cases. Attempts to calculate larger clusters than cyclic pentamers failed and resulted in “Coulomb explosion”. The distance between the OH groups and the positive charge centers of the cations [HO-(CH_2_)_2_Cat^+^] is too short for overcoming the strong repulsive Coulomb forces despite strong cooperative hydrogen bonds. It is interesting to observe that the calculated [HO-(CH_2_)_2_-MIm^+^]*_n_* clusters are lower in energy than the corresponding [HO-(CH_2_)_2_-Py^+^]*_n_* clusters. Due to the acidic C(2)-H position at the imidazolium rings, representing an additional proton donor function, this cation is more polarizable. The relevance of polarizability for the meta-stability of the clusters diminishes with increasing hydroxyl alkyl chain length. For *x* ≥ 3 in the cationic clusters [HO-(CH_2_)*_x_*Cat^+^]*_n_*, the nature of the positively charged centers of the cations becomes negligible. This is why the slopes of the interaction energies are almost similar for larger cluster sizes. In Figure 5, we show that the clusters [HO-(CH_2_)_2_-MIm^+^]*_n_* are an exception and give a smaller slope due to the C(2)-H donor function.

## 5. Role of Hydroxyl Alkyl Chain Length on Energies, Geometries, and Spectroscopic Properties

For demonstrating the role of increasing hydroxyl alkyl chain length, we show the calculated energies per species of the cationic clusters [HO-(CH_2_)*_x_*-Py^+^]*_n_* with *x* = 2–6 and *n* = 2–6 in Figure 6. For all clusters, we took dispersion forces explicitly into account. For [HO-(CH_2_)_2_-Py^+^]*_n_*, the energies increase from 75 kJ mol^−1^ for the dimer up to 250 kJ mol^−1^ for the cyclic pentamer. Extending the alkyl chain by only one methylene group in clusters [HO-(CH_2_)_3_-Py^+^]*_n_* already allows achieving cyclic hexamers with similar energies per cation. If we further lengthen the hydroxyl alkyl chains up to *x* = 6, the energy per cation drops from 250 kJ mol^−1^ down to 160 kJ mol^−1^. We have shown recently that larger clusters than cyclic hexamers can be only achieved if substantially longer hydroxyl octyl groups are considered [59]. To demonstrate this effect, we added the earlier calculated energies for clusters [HO-(CH_2_)_8_-Py^+^]*_n_* in Figure 6. It seems that the critical threshold for meta-stability of the cationic clusters per species is 260 kJ mol^−1^ (see Figure 5 and Figure 6). We tried to calculate cationic clusters with longer hydroxyalkyl chains *x* = 4, 5, 6, but could never optimize structures larger than the cyclic hexamers already obtained for [HO-(CH_2_)_3_-Py^+^]. However, cooperative hydrogen bonding stabilizes the cyclic hexamers, which resembles H-bond structural motifs known from cyclic alkanes or molecular clusters.

The hydroxyl alkyl chain length dependence of [HO-(CH_2_)*_x_*Py^+^]*_n_* clusters is nicely reflected in the H-bond geometries and spectroscopic properties such as NMR ^1^H proton chemical shifts, δ^1^H, and deuteron quadrupole coupling constants, χ_D_. Despite strong electrostatic opposition, the cationic clusters show typical H-bond distances and spectroscopic signatures as known for molecular liquids. The intramolecular bond lengths, *r*_(OH)_, and the intermolecular bond distances, *r*_(O…O)_ of the cationic clusters [HO-(CH_2_)*_x_*-Py^+^]*_n_* are shown in Figure 7 and Figure 8. Stronger hydrogen bonds are reflected in elongated *r*_(OH)_ covalent bonds, and shortened *r*_(O…O)_ hydrogen bonds. It is well-known from the literature that cooperative hydrogen bonding is saturated in cyclic hexamers [60,61]. Larger ring structures are slightly weaker bound as shown here for the intra and inter molecular geometries. Although the cyclic hexamers [HO-(CH_2_)*_x_*-Py^+^]_6_ are meta-stable for *x* = 3–6, the maximum hydrogen bond strength is achieved in the cyclic pentamers. However, for the cyclic hexamer [HO-(CH_2_)_6_-Py^+^]_6_ the geometries only slightly change compared to the pentamer values indicating that repulsive Coulomb interaction is almost compensated by enhanced cooperative hydrogen bonding. Obviously, the weak like-charge repulsion in cationic clusters [HO-(CH_2_)_6_-Py^+^]*_n_* allows the formation of “molecular islands” wherein hydrogen bonds are as strong as in molecular systems. This finding is supported by comparison with intramolecular bond lengths, *r*_(OH)_, and intermolecular bond distances, *r*_(O…O)_ measured by means of neutron and X-ray diffraction for liquid alcohols shown in Figure 7 and Figure 8 [62,63]. This behavior is even better reflected in the calculated spectroscopic properties. The NMR proton chemical shifts, δ^1^H, are plotted for all clusters [HO-(CH_2_)*_x_*-Py^+^]*_n_* versus cluster size *n* as shown in Figure 9. The chemical shielding references are the values of the corresponding monomers leading to δ^1^H values of 0 ppm for each monomer. We observe that the δ^1^H values of the cyclic pentamers and hexamers of [HO-(CH_2_)_5_-Py^+^], and [HO-(CH_2_)_6_-Py^+^] lie in the range of measured proton chemical shifts of liquid methanol recorded between 200 and 300 K. Again, we state that the cationic clusters with sufficiently long hydroxyl alkyl chains at the cation show properties known for molecular clusters and even for the liquid state of alcohols [34,35,36,37,38,46,47,48]. The same holds for deuteron quadrupole coupling constants, χ_D_. In Figure 10, we show the differences χ_D_ between the average cationic cluster values and the deuteron coupling constants of the corresponding monomers. Thus, any negative values indicate hydrogen bonding. For the cyclic pentamers and hexamers of cations [HO-(CH_2_)_5_-Py^+^] and [HO-(CH_2_)_6_-Py^+^], χ_D_ is only slightly smaller than the difference obtained from microwave studies in the gas phase and NMR investigations in the liquid state of methanol [64,65,66,67]. We conclude that cationic clusters with sufficiently long hydroxyl alkyl chains (*x* = 5,6) at the pyridinium cations exhibit NMR spectroscopic properties resembling values in the liquid phase or describing differences between the gas and liquid phase and thus indicating the proper strength of hydrogen bonding.

## 6. Neural Clusters: Competition between (c–a) and (c–c) Hydrogen Bonds

Finally, we calculated neutral complexes ([HO-(CH_2_)_2_-Py][BF_4_])*_n_* with *n* = 1–6 by adding tetrafluoroborate [BF_4_^−^] counter ions to the cationic clusters. In principle, two topologies are possible: simple ion pairs characterized by attractive Coulomb interaction and additional (c–a) hydrogen bonding between cation and anion. One hexamer ([HO-(CH_2_)_2_-Py^+^][BF_4_])_6_ solely built upon six ion pairs is shown in Figure 3b. The other hexamer comprises the pure cyclic cationic cluster with (c–c) hydrogen bonds and six [BF_4_^−^] counter anions that interact with the positively charged rings of the cations as illustrated in Figure 3c. We observed both types of hydrogen bonds in infrared spectra, whereby (c–c) H-bonds are stronger than (c–a) H-bonds and are strongly favored with decreasing temperature. Adding the counterions directly leads to negative binding energies per ion pair [HO-(CH_2_)_2_-Py][BF_4_] for the neutral clusters ([HO-(CH_2_)_2_-Py][BF_4_])*_n_* with *n* = 2–6. As shown in Figure 11, the values converge at −95 kJ mol^−1^ for both types to pentamers and hexamers. Except for the dimers, the (c–a) H-bonded clusters are less than 3 kJ mol^−1^ better in energy than the (c–c) H-bonded clusters. The minor differences in energy explain the experimental observation of both cluster species (c–a) and (c–c) in the IR spectra [20,21,22,23,24]. Both types of H-bonds exist in equilibrium. For the short hydroxyethyl chained cations, the (c–a) H-bonds are favored over (c–c) H-bonds due to the relatively strong repulsive Coulomb forces in the latter. Obviously, cooperative effects seem to be saturated in pentameric and hexameric clusters.

The intramolecular bond lengths, *r*_(OH)_, and the intermolecular bond distances, *r*_(O…O)_ in the (c–a) and (c–c) H-bonded neutral clusters ([HO-(CH_2_)_2_-Py][BF_4_])*_n_* are shown in Figure 12 and Figure 13 along with the values of the purely cationic clusters [HO-(CH_2_)_2_-Py^+^]*_n_*. The interesting finding is that the (c–a) H-bonded OH groups are not elongated but shortened with increasing cluster size. The strong H-bond in the isolated ion pair monomer is rather weakened than strengthened upon dissolving in the environment of other ion pairs. In contrast, the OH-bonds in the (c–c) bound clusters are elongated due to increasing cooperative effects in the cyclic structures, although less enhanced with increasing cluster size as in the pure cationic clusters.

We report similar observations for the intermolecular bond distances, *r*_(O…O)_, that are shortened with increasing size for the (c–c) clusters. There is one exception, which is the (c–c) H-bonded trimer. The cyclic (c–c) trimer does not survive the geometry optimization and is reformed into a linear trimer with two (c–c) H-bonds and one (c–a) H-bond where the final OH group interacts with one of the anions. This linear trimer is characterized by less cooperativity resulting in significantly weaker binding and longer H-bond distances *r*_(O…O)_.

The hydroxyl alkyl chain length dependence of neutral clusters ([HO-(CH_2_)_2_-Py][BF_4_])*_n_* is better reflected in the NMR δ^1^H proton chemical shifts of the hydroxyl groups, which are sensitive probes of hydrogen bonding. In Figure 14, we show the δ^1^H values for the neutral (c–a) and (c–c) H-bonded clusters ([HO-(CH_2_)_2_-Py][BF_4_])*_n_*, all with reference to the value of the isolated (c–a) ion pair. In accord with the H-bond geometries, the proton chemical shifts are upfield shifted for (c–a) clusters and downfield shifted for (c–c) clusters with increasing clusters size. We explained the exceptional value for the (c–c) trimer already above. The geometry optimization results in a linear trimer rather than in a cyclic structure. The Δ(δ^1^H) difference of the proton chemical shifts between (c–a) and (c–c) H-bonded tetramers, pentamers and hexamers lies in the range of 2.5 to 3.5 ppm. Thus, if fast proton exchange in the ILs can be suppressed at a very low temperature, both NMR signals should be observable separately. Similar is true for the related deuteron quadrupole coupling constants providing the same sensitivity for hydrogen bonding as the proton chemical shifts. As shown in Figure 15, the differences Δχ_D_ between the values of the large (c–a) and (c–c) cluster species are in the order of 35 kHz. In recent NMR solid state experiments on ILs [HO-(CH_2_)_2_-Py][NTf_2_] and [HO-(CH_2_)_2_-Py][NTf_2_] near the glass transition at 183 K we measured Δ(χ_D_) = 41 kHz and Δ(χ_D_) = 50 kHz, respectively [25,27]. Here, we show that the calculated spectroscopic properties of the clusters are similar to those observed in the condensed phase suggesting that these structural motifs are present in the glassy ionic liquids.

## 7. Universal Linear Relation between Δ(χ_D_) and δ^1^H Allows Prediction of Properties

Finally, we show that the earlier presented universal relation between deuteron quadrupole coupling constant differences, Δ(*χ*_D_) and the corresponding proton chemical shifts, *δ*^1^H also holds for the (c–a) and (c–c) hydrogen bonded clusters [68]. For the cationic clusters, spectroscopic monomer (gas phase) values are taken as reference (Δ(*χ*_D_) = *δ*^1^H = 0). For the neutral clusters ([HO-(CH_2_)_2_-Py][BF_4_])*_n_* (c–a) and ([HO-(CH_2_)_2_-Py][BF_4_])*_n_* (c–c) the isolated ion pair monomer values are used as reference. The linear relation between the NMR spectroscopic properties is shown in Figure 16. No matter whether we consider cationic clusters [HO-(CH_2_)_2_-Py^+^]*_n_*, [HO-(CH_2_)_3_-Py^+^]*_n_*, and [HO-(CH_2_)_2_-Pip^+^]*_n_*, with differently polarizable cations or varying hydroxyl chain lengths or if we take neutral clusters ([HO-(CH_2_)_2_-Py][BF_4_])*_n_* (c–a) and ([HO-(CH_2_)_2_-Py][BF_4_])*_n_* (c–c) into account, all pairs of NMR spectroscopic properties are described by linear behavior. The weakly bound O-H groups in clusters [HO-(CH_2_)_2_-Pip^+^]*_n_* are characterized by small *δ*^1^H-shifts and slightly negative (*χ*_D_) values. Clusters including the more polarizable cation [HO-(CH_2_)_2_-Py^+^] exhibit large chemical shifts corresponding increased negative (*χ*_D_) values, both indicating stronger hydrogen bonding. Increasing the alkyl chain length of the pyridinium cation results in further NMR downfield chemical shifts and larger deuteron quadrupole coupling constant differences due to weakening the repulsive Coulomb forces and enhancing the hydrogen bonds at the same time. It does not matter whether these pairs of properties are calculated for O-H/O-D in varying sized clusters or for different configurations within these clusters; they all show linear dependence. Obviously, both properties, *δ*^1^H and (*χ*_D_) are similarly sensitive to local and directional interactions, such as hydrogen bonding. The universal relation allows predicting the rarely known deuteron quadrupole coupling constants for ionic liquids by measuring the easily accessible proton chemical shifts [64,68].

## 8. Conclusions

In this work, we show that ring clusters of monovalent hydroxyl-functionalized cations [HO-(CH_2_)*_x_*Cat^+^]*_n_* with *x* = 2–6 and *n* = 3–6 are kinetically stable depending on the polarizability of the cations and in particular on the hydroxyl alkyl chain length. Cooperative hydrogen bonding prevents the intrinsically meta-stable cyclic clusters from “Coulomb explosion”. For the short chained hydroxethyl cations [HO-(CH_2_)_2_Cat^+^]*_n_*, the polarizability of the cations is important for the achievable size of meta-stable clusters. Non-polarizable cations such as hydroxyethyl ammonium allow the formation of linear dimers only, whereas polarizable cations such as hydroxyethyl pyridinium allow for building meta-stable cyclic tetramers. If we take dispersion forces explicitly into account, the cluster sizes increase to cyclic tetramers in the first case and to cyclic pentamers in the latter case. For the cationic clusters [HO-(CH_2_)*_x_*Py^+^]*_n_* with *x* = 2–6 and *n* = 2–6 we studied the relevance of the hydroxyl alkyl chain length dependence for the meta-stability of the clusters. The long hydroxyalkyl chains increase the distance between the positive charge centers at the pyridinium ring and the hydroxyl groups, resulting in significantly reduced Coulomb interaction and enhanced cooperative hydrogen bonding. Larger clusters than cyclic hexamers are not achievable, even if dispersion interaction is considered. There seems to be a meta-stability limit. Higher energies per species than 260 kJ mol^−1^ lead to “Coulomb explosion”. However, geometries and spectroscopic properties of the largest cationic clusters resemble those of similar sized molecular clusters of alcohols showing that the H-bond motifs are almost untouched by repulsive Coulomb forces. The neutral clusters ([HO-(CH_2_)_2_Py][BF_4_])*_n_* exhibiting both (c–a) or (c–c) hydrogen bonds, are close in energy, reflecting IR spectroscopic measurements showing increasing contributions of the (c–c) vibrational bands with decreasing temperature. We conclude that pure cationic clusters provide robust meta-stability, whereas cationic clusters in overall neutral bulk-like environment successfully compete with similar sized clusters comprising ion pairs (c–a) solely. We also show that the structural and spectroscopic features of calculated clusters resemble those observed in liquid phase experiments.

## Figures and Tables

**Figure 1 molecules-25-04972-f001:**
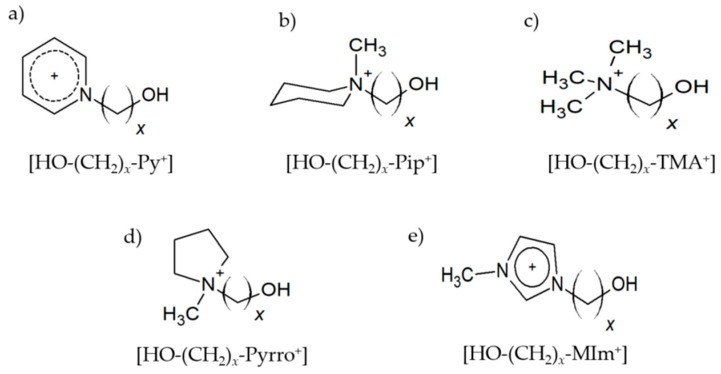
Hydroxy-functionalized cations (**a**) 1-(*x*-hydroxyalkyl)pyridinium [HO-(CH_2_)*_x_*-Py^+^], (**b**) 1-(*x*-hydroxyalkyl)methylpiperidinium [HO-(CH_2_)*_x_*-Pip^+^], (**c**) 1-(*x*-hydroxyalkyl)-1,1,1-trimethylammonium [HO-(CH_2_)*_x_*-TMA^+^], (**d**) 1-(*x*-hydroxyalkyl)methylpyrrolidinium [HO-(CH_2_)*_x_*-Pyrro^+^], and (**e**) 1-(*x*-hydroxyalkyl)-3-methylimidazolium [HO-(CH_2_)*_x_*-MIm^+^] with varying chain length *x* = 2–6 as constituents of cationic clusters.

**Figure 2 molecules-25-04972-f002:**
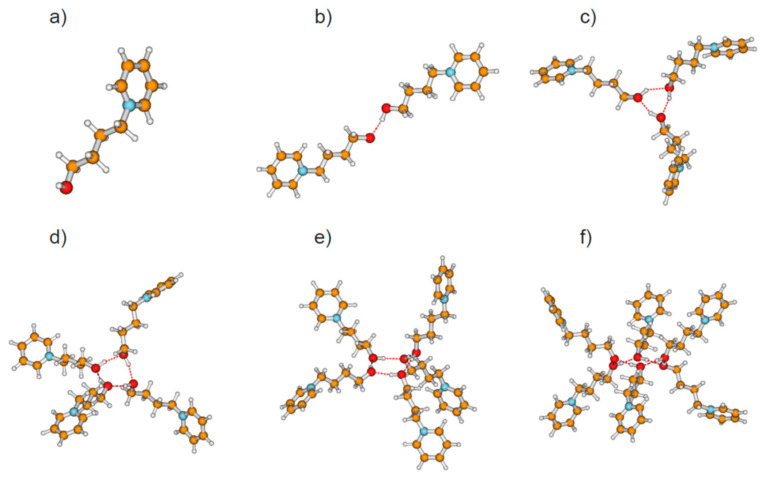
Topologies of calculated cationic clusters [HO-(CH_2_)_4_-Py^+^]*_n_* with *n* = 1–6 comprising (**a**) monomers, (**b**) linear dimers, (**c**) cyclic trimers, (**d**) cyclic tetramers, (**e**) cyclic pentamers, and (**f**) cyclic hexamers as obtained from density functional theory (DFT) calculations at the B3LYP-D3/6–31+G* level of theory.

**Figure 3 molecules-25-04972-f003:**
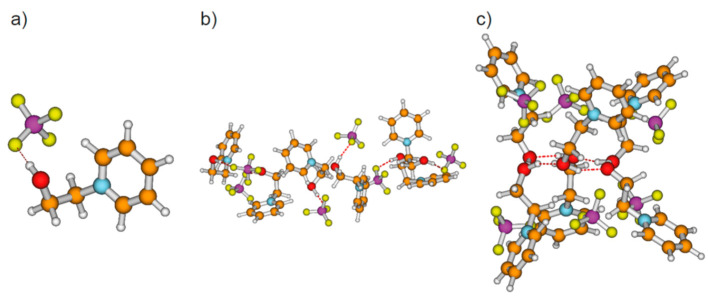
(**a**) Simple ion pair with (c–a) hydrogen bond, (**b**) neutral hexameric cluster ([HO-(CH_2_)_2_-Py^+^][BF_4_])_6_ comprising six ion pairs with (c–a) hydrogen bonds solely, and (**c**) neutral hexameric cluster ([HO-(CH_2_)_2_-Py^+^][BF_4_])_6_ characterized by the cooperative (c–c) H-bonded hexameric ring with six [BF_4_^−^] anions, each interacting with the positive charge at the pyridinium rings of the cations.

**Figure 4 molecules-25-04972-f004:**
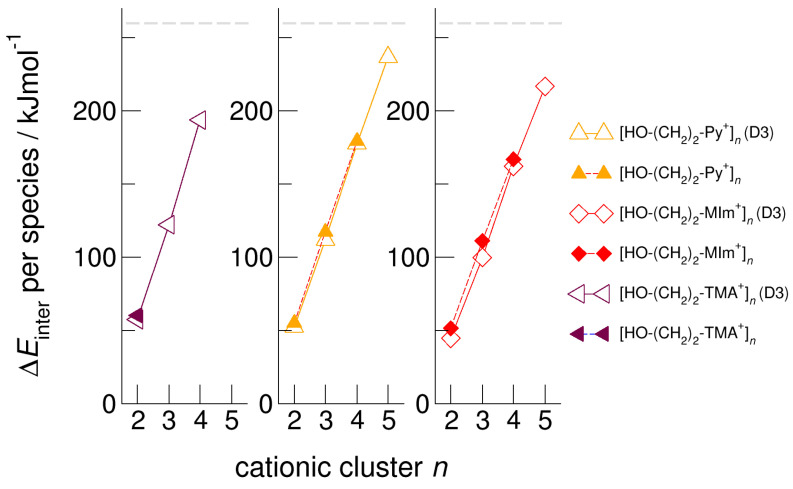
Intermolecular energies per species in the cationic clusters [HO-(CH_2_)_2_-TMA^+^]*_n_*, [HO-(CH_2_)_2_-Py^+^]*_n_*, and [HO-(CH_2_)_2_-MIm^+^]*_n_* calculated at the B3LYP/6–31+G* level of theory with (open symbols) and without (filled symbols) dispersion correction. The dashed grey lines indicate meta-stability of the cationic clusters with energies below 260 kJ mol^−1^ per cation. Beyond this energy, the cationic clusters undergo “Coulomb explosion” and dissociate [59].

**Figure 5 molecules-25-04972-f005:**
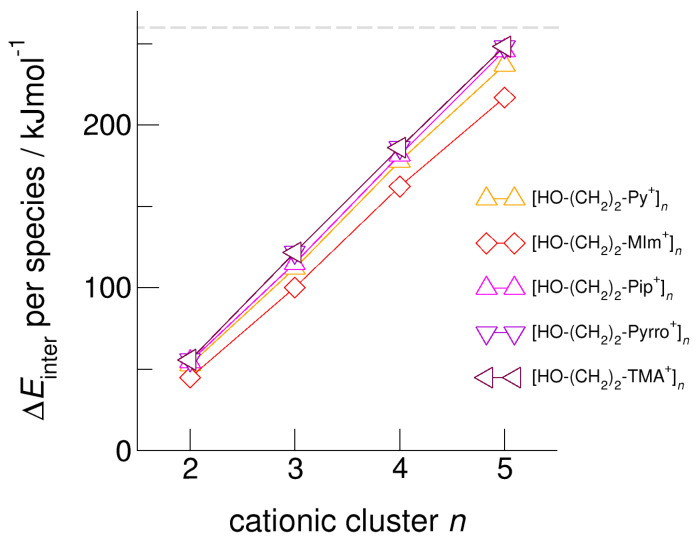
Intermolecular energies per species in the cationic clusters [HO-(CH_2_)_2_-Py^+^]*_n_*, [HO-(CH_2_)_2_-MIm^+^]*_n_*, [HO-(CH_2_)_2_-Pip^+^]*_n_*, [HO-(CH_2_)_2_-Pyrro^+^]*_n_*, and [HO-(CH_2_)_2_-TMA^+^]*_n_* calculated at the B3LYP-D3/6–31+G* level of theory. The dashed grey line indicates meta-stability of the cationic clusters with energies below 260 kJ mol^−1^ per cation. Beyond this energy, the cationic clusters undergo “Coulomb explosion” and dissociate [59].

**Figure 6 molecules-25-04972-f006:**
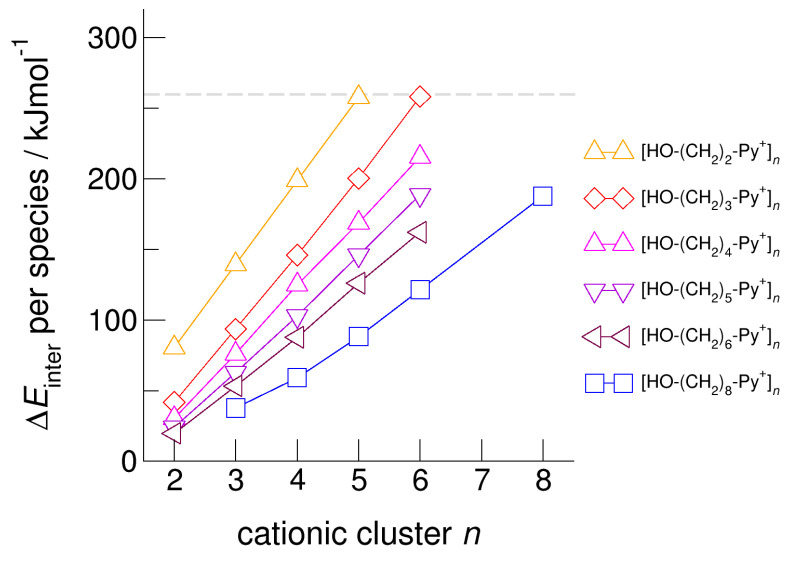
Intermolecular energies per species in the cationic clusters [HO-(CH_2_)*_x_*-Py^+^]*_n_* with *x* = 2–6 calculated at the B3LYP-D3/6–31+G* level of theory. The dashed grey line indicates meta-stability of the cationic clusters with energies below 260 kJ mol^−1^ per cation. Beyond this energy, the cationic clusters undergo “Coulomb explosion” and dissociate [53]. We also added recently calculated energies for cationic clusters [HO-(CH_2_)_8_-Py^+^]*_n_* [59].

**Figure 7 molecules-25-04972-f007:**
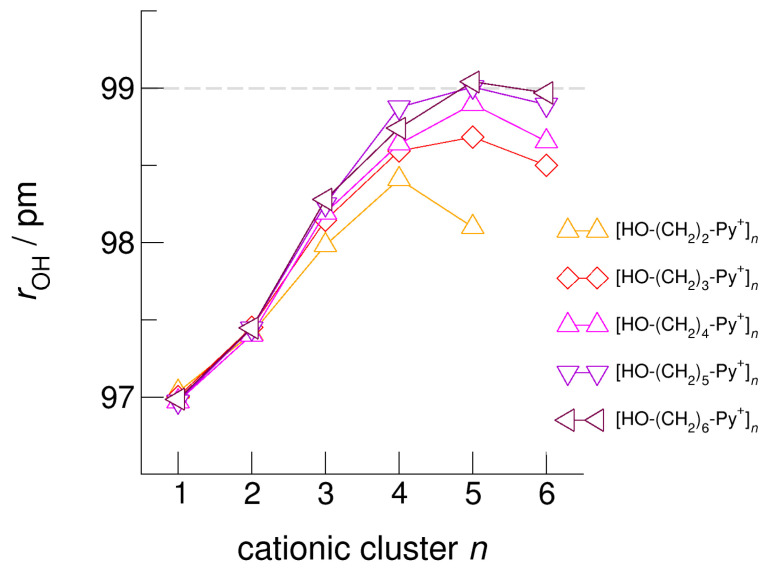
Average intramolecular bond lengths, *r*_OH_, for the cationic clusters [HO-(CH_2_)*_x_*-Py^+^]*_n_* with *x* = 2–6 and *n* = 2–6. For the clusters [HO-(CH_2_)_6_-Py^+^]*_n_* with the longest hydroxyalkyl chains, *r*_OH_, is elongated by 2 ppm due to enhanced cooperative hydrogen bonding. For comparison, we show the *r*_OH_ bond distance of about 99 ± 1 pm measured for liquid methanol by means of neutron diffraction data (grey dashed line) [62].

**Figure 8 molecules-25-04972-f008:**
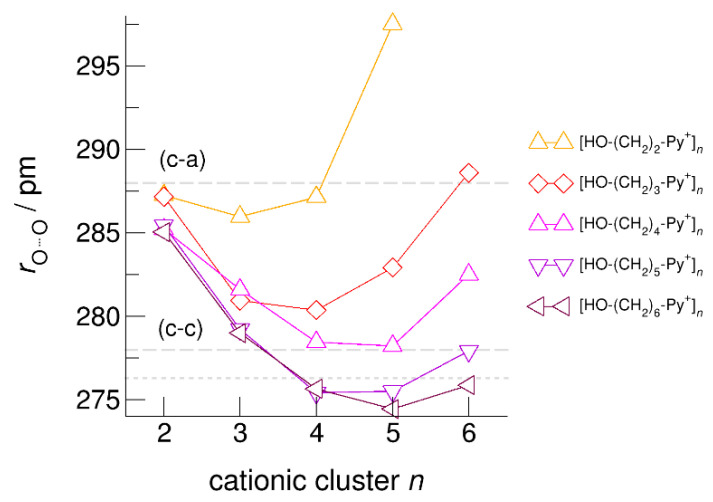
Average intermolecular bond lengths, *r*_(O…O)_, for the cationic clusters [HO-(CH_2_)*_x_*-Py^+^]*_n_* with *x* = 2–6 and *n* = 2–6. For the clusters [HO-(CH_2_)_6_-Py^+^]*_n_* with the longest hydroxyalkyl chains, *r*_(O…O)_ is shortened by 10 ppm due to enhanced cooperative hydrogen bonding. For comparison, we show the average *r*_(O…O)_ distances of the pure ionic liquid [HO-(CH_2_)_4_-Py][NTf_2_] obtained from neutron diffraction (grey dashed lines) along with those for the ring hexamer of liquid ethanol derived from X-ray diffraction data (grey dotted line) [24,63].

**Figure 9 molecules-25-04972-f009:**
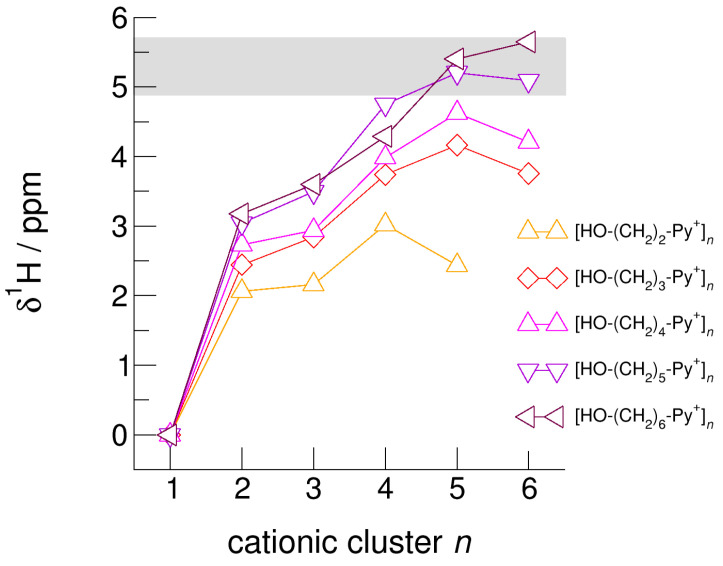
Calculated average NMR proton chemical shifts, *δ*^1^H, of the hydroxyl protons for the cationic clusters [HO-(CH_2_)_2_-Py^+^]*_n_*, [HO-(CH_2_)_3_-Py^+^]*_n_*, [HO-(CH_2_)_4_-Py^+^]*_n_*, [HO-(CH_2_)_5_-Py^+^]*_n_*, and [HO-(CH_2_)_6_-Py^+^]*_n_* related to the corresponding monomer values. For comparison, we added the *δ*^1^H OH chemical shifts of liquid methanol in the temperature range between 200 K (*δ*^1^H = 5.7 ppm) and 300 K (*δ*^1^H = 4.9 ppm) as indicated by the grey area [64].

**Figure 10 molecules-25-04972-f010:**
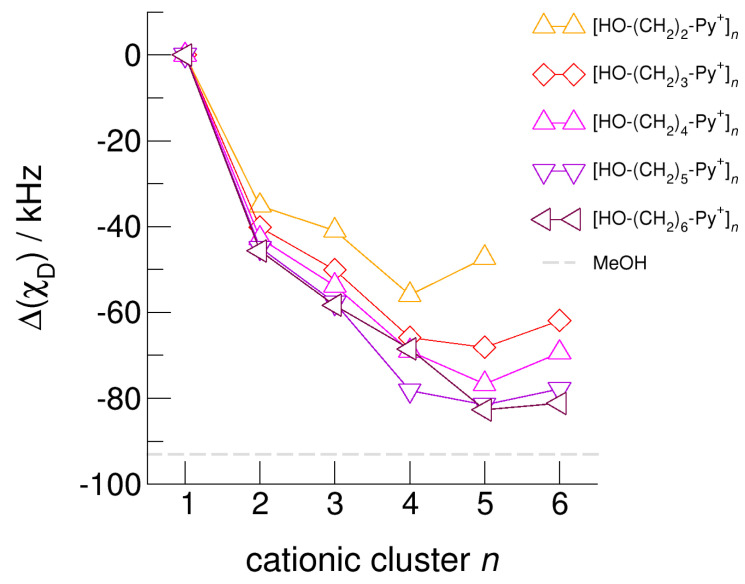
Calculated average deuteron quadrupole coupling constants, *χ*_D_, of the hydroxyl deuterons for the cationic clusters [HO-(CH_2_)_2_-Py^+^]*_n_*, [HO-(CH_2_)_3_-Py^+^]*_n_*, [HO-(CH_2_)_4_-Py^+^]*_n_*, [HO-(CH_2_)_5_-Py^+^]*_n_*, and [HO-(CH_2_)_6_-Py^+^]*_n_* related to the corresponding monomer values. For comparison, we show the (χ_D_) difference of deuteron quadrupole coupling constants of methanol MeOH in the liquid and the gas phase (grey dashed line) [65,66,67].

**Figure 11 molecules-25-04972-f011:**
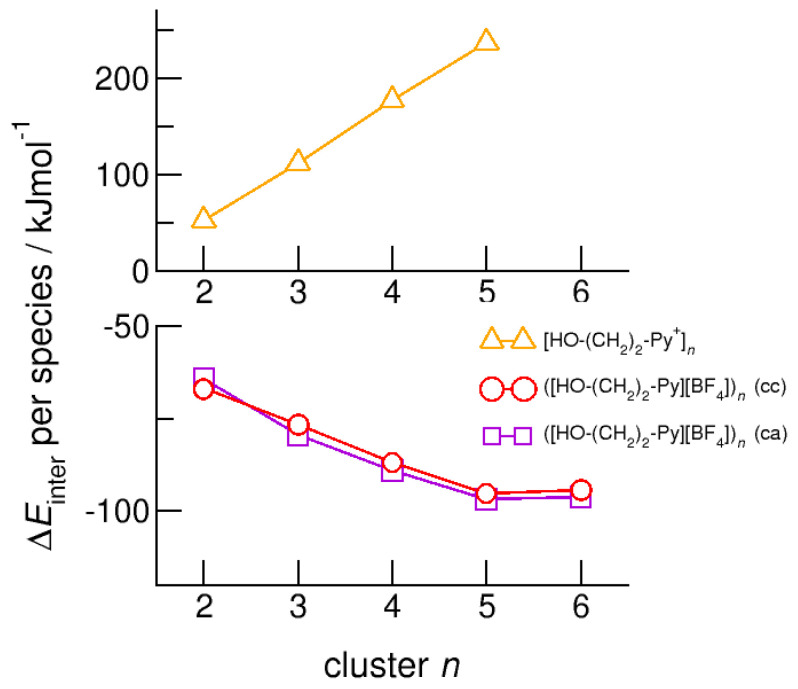
Intermolecular energies per species in the cationic clusters [HO-(CH_2_)_2_-Py^+^]*_n_* (triangles), ([HO-(CH_2_)_2_-Py][BF_4_])*_n_* (c–c) (circles) and ([HO-(CH_2_)_2_-Py][BF_4_])*_n_* (c–a) (squares) calculated at the B3LYP-D3/6–31+G* level of theory.

**Figure 12 molecules-25-04972-f012:**
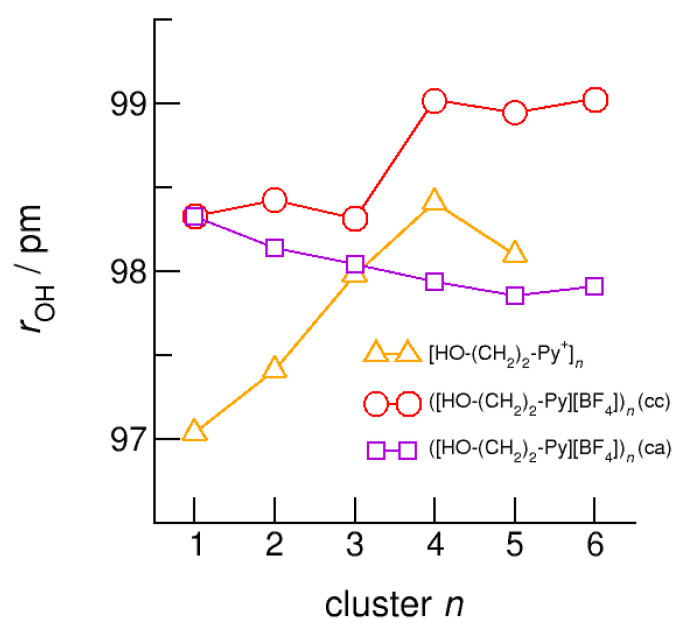
Average intramolecular bond lengths, *r*(OH), for the cationic clusters [HO-(CH_2_)_2_-Py^+^]*_n_* (triangles), ([HO-(CH_2_)_2_-Py][BF_4_])*_n_* (c–c) (circles) and ([HO-(CH_2_)_2_-Py][BF_4_])*_n_* (c–a) (squares) calculated at the B3LYP-D3/6–31+G* level of theory.

**Figure 13 molecules-25-04972-f013:**
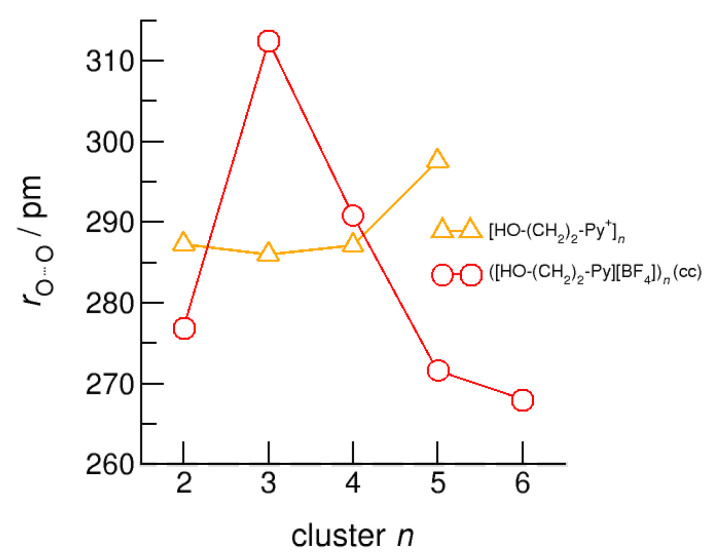
Average intermolecular bond lengths, *r*(O…O), for the cationic clusters [HO-(CH_2_)_2_-Py^+^]*_n_* (triangles) and neutral clusters ([HO-(CH_2_)_2_-Py][BF_4_])*_n_* (c–c) (circles) calculated at the B3LYP-D3/6–31+G* level of theory [24].

**Figure 14 molecules-25-04972-f014:**
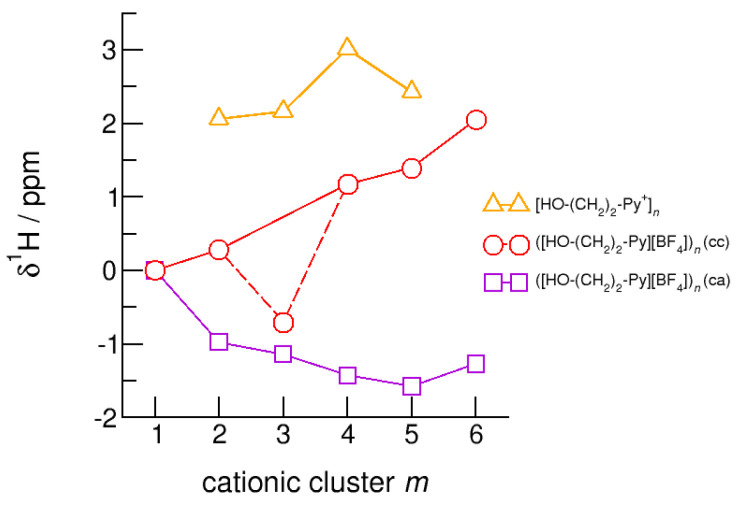
Average NMR proton chemical shifts, *δ*^1^H, of the hydroxyl protons for the cationic clusters [HO-(CH_2_)_2_-Py^+^]*_n_* (triangles), ([HO-(CH_2_)_2_-Py][BF_4_])*_n_* (c–c) (circles) and ([HO-(CH_2_)_2_-Py][BF_4_])*_n_* (c–a) (squares) related to the corresponding monomer values calculated at the B3LYP-D3/6–31+G* level of theory.

**Figure 15 molecules-25-04972-f015:**
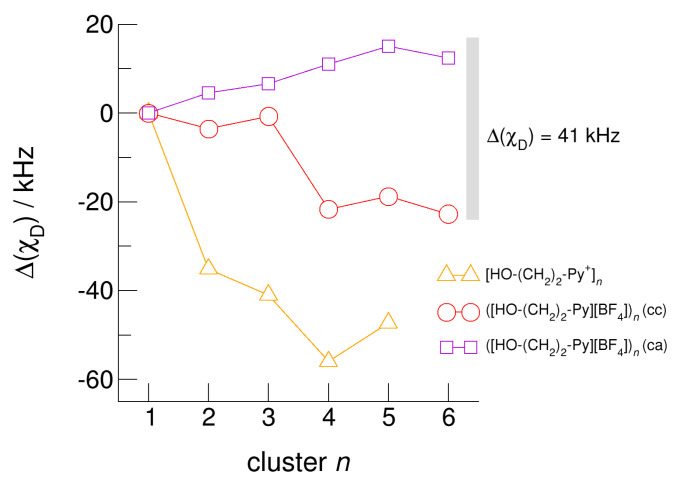
Average deuteron quadrupole coupling constants, *χ*_D_, of the hydroxyl deuterons for the cationic clusters [HO-(CH_2_)_2_-Py^+^]*_n_* (triangles), the neutral clusters ([HO-(CH_2_)_2_-Py][BF_4_])*_n_* (c–c) (circles) and ([HO-(CH_2_)_2_-Py][BF_4_])*_n_* (c–a) (squares) related to the corresponding monomer values calculated at the B3LYP-D3/6–31+G* level of theory. The grey bar indicates the Δ(*χ*_D_) = 41 kHz difference between the (c–c) and (c–a) hydrogen bonds in the ionic liquid [HO-(CH_2_)_2_-Py][NTf_2_] as observed in NMR solid state experiments at the glass transition [25].

**Figure 16 molecules-25-04972-f016:**
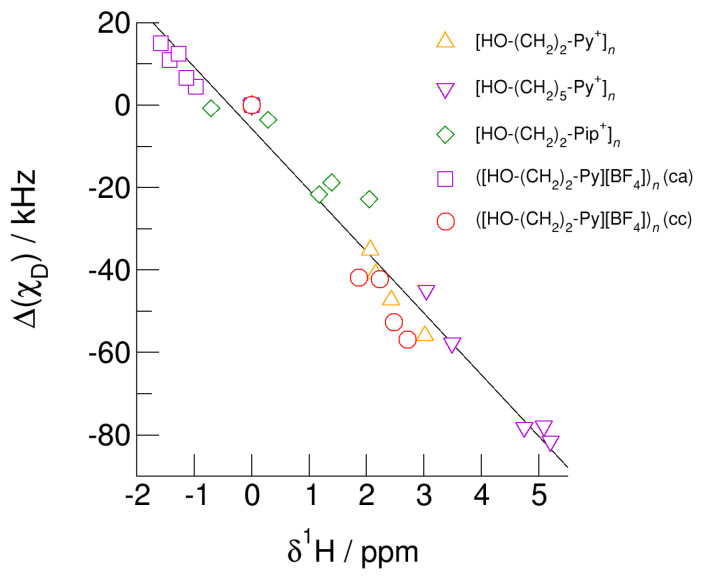
Relation between deuteron quadrupole coupling constant differences, *(χ*_D_), of the hydroxyl deuterons and proton chemical shifts, *δ*^1^H, of the hydroxyl protons in cationic clusters [HO-(CH_2_)_2_-Py^+^]*_n_* (triangles up), [HO-(CH_2_)_5_-Py^+^]*_n_* (triangles down), [HO-(CH_2_)_2_-Pip^+^]*_n_* (diamonds), ([HO-(CH_2_)_2_-Py][BF_4_])*_n_* (c–a) (squares), and ([HO-(CH_2_)_2_-Py][BF_4_])*_n_* (c–c) (circles) each compared to the monomer values. Whether we use different cations and alkyl chains in the cationic clusters or (c–a) and (c–c) hydrogen bonding in the neutral clusters ([HO-(CH_2_)_2_-Py][BF_4_])*_n_* (c–c) and ([HO-(CH_2_)_2_-Py][BF_4_])*_n_* (c–a), the spectroscopic properties show a universal linear behavior. Easy access to NMR proton chemical shifts allows reliably predicting deuteron quadrupole coupling constants that are usually unknown for clusters and the bulk liquid phase.

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
