# Peer review of "Clusters of Hydroxyl-Functionalized Cations Stabilized by Cooperative Hydrogen Bonds: The Role of Polarizability and Alkyl Chain Length"

_molecules, 2020, doi:10.3390/molecules25214972_

Round 1
Reviewer 1 Report
In my opinion the submitted work entitled “Clusters of hydroxyl-functionalized cations stabilized by cooperative hydrogen bonds: The role of polarizability and alkyl chain length” after introducing the corrections listed below, may be published..
I think that in the caption for Figure 1 one should assign the appropriate cations, as shown in the figure, to the symbols a) ... e).
There is no reference (citation) in the text to Fig. 15.
Line 174 – please explain DQCC abbreviation or enter this abbreviation on line 165 for “deuteron quadrupole coupling constants” name.
Literature:
[36] please add the end pages of the publication - should be J. Chem. Phys. 1998, 108, 20-32
[38] page numbers for ‘Phys. Chem. Chem. Phys. 2018, 20, 29184-20206’ should be replaced by 29184-29206
Author Response
Review 1:
Comments and Suggestions for Authors
In my opinion the submitted work entitled “Clusters of hydroxyl-functionalized cations stabilized by cooperative hydrogen bonds: The role of polarizability and alkyl chain length” after introducing the corrections listed below, may be published..
I think that in the caption for Figure 1 one should assign the appropriate cations, as shown in the figure, to the symbols a) ... e).
A: We added the symbols in the figure caption.
There is no reference (citation) in the text to Fig. 15.
A: We now reference Fig. 15 in the text.
Line 174 – please explain DQCC abbreviation or enter this abbreviation on line 165 for “deuteron quadrupole coupling constants” name.
A: We introduced the DQCC abbreviation on p.5, l. 112
Literature:
[36] please add the end pages of the publication - should be J. Chem. Phys. 1998, 108, 20-32
[38] page numbers for ‘Phys. Chem. Chem. Phys. 2018, 20, 29184-20206’ should be replaced by 29184-29206
A: We added and corrected the page numbers in both references.
Reviewer 2 Report
The paper describes quantum chemical calculations for studying clusters of hydroxyl functionalized cations kinetically stabilized by hydrogen bonding and the role of polarizability and alkyl chain length. The paper could be published subject to several revisions as indicated below:
- Page 5, lines 97-98: “…electrostatic forces between the like-charged ions challenge the generally accepted electrostatic model for hydrogen bonds (HBs)”.
For hydrogen bonds, the generally accepted model is not the purely electrostatic model but the one that includes also the importance of covalency in hydrogen bonding (J. Chem. Phys. 1983, 78, 4066–4073; Mol. Phys. 2012, 110, 565–579; Chem. Rev. 2011, 111, 2597–2625).
- Page 8, lines 167-169: “We also calculated two spectroscopic descriptors, the hydroxyl proton NMR chemical shifts, 1H, and the deuteron quadrupole coupling constants, χD, which are both sensitive probes for hydrogen bonding.”.
The hydroxyl 1H NMR chemical shifts (both phenol and alcohol type) are by far the most sensitive tool which are easily determined experimentally and computationally. Reference should be given to several recent articles in the field of DFT calculations of phenol OH and alcohol OH 1H NMR chemical shifts.
- Page 22, lines 413-414.
“Finally, we present a universal relation between deuteron quadrupole coupling constant differences, ( D) and the corresponding proton chemical shifts, 1H”.
A relationship between deuteron quadrupole coupling constant differences, (D) and the corresponding proton chemical shifts is already reported in Phys. Chem. Chem. Phys. 2016, 18, 17788-17794 (Fig.1).
- Figure 7.
“Average intramolecular bond lengths, rOH, for the cationic clusters [HO-(CH2)x-Py+ 303 ]n with x=2-6 and n=2-6. For the clusters [HO-(CH2)6-Py+]n with the longest hydroxyalkyl chains, rOH, is elongated by 2 ppm due to enhanced cooperative hydrogen bonding. For comparison, we show the rOH bond distance of liquid methanol derived from neutron diffraction data (grey dashed line)]”.
It should be emphasized that the accuracy in determining OH distances by neutron diffraction has been reported to be 0.990 ±0.010 Å which is, in fact, the whole range of the vertical scale of Fig.7 of the present manuscript. Accuracy, therefore, and physical meaning of Fig. 7 should not be exaggerated.
- Figure 8.
“Average intermolecular bond lengths, r(O…O), for the cationic clusters [HO-(CH2)x-Py+]n with x=2-6 and n=2-6. For the clusters [HO-(CH2)6-Py+ ]n with the longest hydroxyalkyl chains, r(O… O) is shortened by 10 ppm due to enhanced cooperative hydrogen bonding. For comparison we show the r(O…O) distance for the ring hexamer of liquid ethanol derived from x-ray diffraction data (grey dashed line)”.
Comparison of the hypothetical ring hexamer of liquid ethanol derived from x-ray diffraction has two disadvantages: (i) the two systems are completely different and (ii) analysis of X-ray diffraction data of liquid ethanol at room temperatures has a high degree of uncertainty. For example, the OH distance of 0.890 Å is extremely short. See Cryst. Growth Des. 2012, 12, 1014-1021; Molecules 2017, 22, 415.
- Figure 9.
“Calculated average NMR proton chemical shifts, 1H, of the hydroxyl protons for the cationic clusters [HO-(CH2)2-Py+ ]n, [HO-(CH2)3-Py+ ]n, [HO-(CH2)4-Py+ ]n, [HO-(CH2)5-Py+]n, and [HO-(CH2)6-Py+ ]n related to the corresponding monomer values. For comparison, we added the 1H OH chemical shifts of liquid methanol in the temperature range between 200 K (1H=5.7 ppm) and 300 K (1H=4.9 ppm) as indicated by the grey area).”.
The chemical shifts of liquid methanol in the temperature range between 200 K (1H=5.7 ppm) and 300 K (1H=4.9 ppm) is probably not the best model for comparison with 1H NMR chemical shifts of hydroxyl protons for cationic clusters.
- Figure 10.
“Calculated average deuteron quadrupole coupling constants, D, of the hydroxyl deuterons for the cationic clusters [HO-(CH2)2-Py+ ]n, [HO-(CH2)3-Py+ ]n, [HO-(CH2)4-Py+]n, [HO-(CH2)5- Py+ ]n, and [HO-(CH2)6-Py+]n related to the corresponding monomer values. For comparison, we show the ( D) difference of deuteron quadrupole coupling constants of methanol MeOH in the liquid and the gas phase (grey dashed line).”.
The same comments as above.
